# The Impact of the COVID-19 Pandemic and Associated Societal Restrictions on People Experiencing Homelessness (PEH): A Qualitative Interview Study with PEH and Service Providers in the UK

**DOI:** 10.3390/ijerph192315526

**Published:** 2022-11-23

**Authors:** Jo Dawes, Tom May, Daisy Fancourt, Alexandra Burton

**Affiliations:** 1Collaborative Centre for Inclusion Health, Institute of Epidemiology and Health Care, University College London (UCL), London WC1E 7HB, UK; 2Department of Behavioural Science and Health, Institute of Epidemiology and Health Care, University College London (UCL), London WC1E 7HB, UK

**Keywords:** homelessness, COVID-19, health, housing, support services, coping, pandemic, qualitative

## Abstract

People experiencing homelessness (PEH) faced unique challenges during the COVID-19 pandemic, including changes to accommodation availability, societal restrictions impacting access to essentials like food, and services moving to online and remote access. This in-depth qualitative research aims to add to the existing, but limited research exploring how the pandemic affected PEH. 33 semi-structured qualitative interviews (22 with PEH during the pandemic and 11 with homelessness sector service providers) were undertaken in the United Kingdom between April 2021 and January 2022. Interviews were audio-recorded, transcribed and analysed using reflexive thematic analysis. To ensure consistency of coding, 10% of interviews were coded by two researchers. The PEH sample was 50% female, aged 24–59 years, 59% white British, and included people who had lived in hostels/hotels, with friends/family, and on the streets during the COVID-19 pandemic. Providers came from varied services, including support charities, housing, and addiction services. Five key themes were identified: (i) the understanding of and adherence to public health guidance and restrictions; (ii) the experience of people accommodated by the ‘Everyone In’ initiative; (iii) the impact of social distancing guidelines on PEH experiences in public spaces; (iv) the importance of social support and connections to others; and (v) how homelessness services adapted their provision. Policy makers and public health communicators must learn from PEH to maximize the effectiveness of future public health strategies. Housing providers and support services should recognize the implications of imposing a lack of choice on people who need accommodation during a public health emergency. The loss of usual support for PEH triggered a loss of ability to rely on usual ‘survival strategies’, which negatively influenced their health. This research highlights successes and difficulties in supporting PEH during the COVID-19 pandemic and informs planning for similar public health events.

## 1. Introduction

As the SARS-CoV-2 virus swept the globe, the World Health Organisation declared the outbreak as a pandemic on 12 March 2020 [1]. In response, governments worldwide applied a range of population-level restrictions in an attempt to limit virus transmission. In the United Kingdom (UK) restrictions included constraints to movement, self-isolation, and the closure of non-essential businesses, workplaces, hospitality, and places of education. Whilst the purpose of these measures was to control the virus, the consequences for many were substantial societal disruption, with many people reporting psychological distress, social isolation, and economic hardship [2,3].

One particular restriction imposed in the UK, initiated on 23 March 2020, was the guidance of “staying at home and away from others (social distancing)” [4]. This brought the social disparity of homelessness into acute focus by exposing the impossibility for people without a home to follow it. Moreover, government guidance recommended that communal shelters for people experiencing homelessness (PEH) either be closed or transformed into self-contained accommodation to reduce the spread of COVID-19 [5]. A modelling study suggests that these adapted services provided protection from COVID-19 and saved lives [6]. Subsequently, research carried out in Atlanta, Georgia, USA, demonstrated the positive impact of establishing isolation and non-congregate hotels on COVID-19 symptom evolution in PEH during the pandemic [7]. However, closure of many sites also reduced accommodation provision.

It is known that PEH have substantially poorer health than the general population, with cohort studies showing a three- to six-fold increased mortality risk, high levels of chronic illness and mental health problems [8,9]. In recognition that homelessness during a pandemic potentially carried additional health risks, the UK Government attempted to limit homelessness by adopting policies, including temporarily suspending evictions for 90 days from March 2020, restarting possession orders, extending notice periods before eviction could be started, and suspending bailiff powers to evict [10]. Additionally, from the 26th March they funded the GBP 3.2 million ‘Everyone In’ campaign, to help people who were sleeping on the street to self-isolate, by providing temporary emergency accommodation and supporting appropriate agencies to facilitate this [11]. Prior to the pandemic, the number of people sleeping on the streets in 2019 was estimated at 4266 [12]. However, by January 2021 The Ministry of Housing, Communities and Local Government Select Committee reported the number of people helped by the scheme had risen to 37,500 [13], suggesting a huge underestimation of street homelessness.

Extensive research has been carried out to understand the impact of the pandemic and related restrictions on society, but there is limited literature on understanding how excluded and seldom heard groups, who are often under-represented in research [14], were affected. Changes in access to primary health care services and their impact on care for PEH has been recognized as an important topic, with publication of a study protocol which sets out to examine this, but the full findings are not yet in the public domain [15]. In the UK, qualitative research has been carried out exploring the response of one homeless service in Scotland [16], access to mental health and substance use support among PEH living in the North East of England [17], and community perspectives about why COVID-19 infections were lower than expected in London during 2020 [18]. These studies, which illuminate local perspectives, provide an extremely useful basis from which our research intends to build.

In recognition that experiences can differ by region, our qualitative research sought a national view, by asking the research question: ‘how did the COVID-19 pandemic and associated social distancing restrictions impact the health, wellbeing, and provision of services for PEH in the UK?’ We sought to answer this by collecting and analysing the reported experiences of PEH and staff working with PEH during the COVID-19 pandemic.

## 2. Materials and Methods

We conducted a semi-structured qualitative interview study in the UK with PEH during the COVID-19 pandemic and homelessness sector service providers. The study formed part of the COVID-19 Social Study [19,20]. This study was performed in accordance with the Declaration of Helsinki. Ethical approval was granted by the University College London research ethics committee (Project ID: 6357/002).

### 2.1. Sample and Recruitment

Homelessness has a variety of definitions depending on whether it is understood as a housing, social welfare, or a psychosocial problem. For the purposes of this research, we defined PEH to include the definitions of ‘roofless’ (e.g., no fixed abode, living in a public space) and ‘houseless’ (e.g., living in hostel, refuge, shelter, temporary accommodation) as defined by the European Federation of National Organisations Working with the Homeless (FEANTSA) [21].

We recruited two groups of participants, namely PEH during the COVID-19 pandemic and service providers working with PEH at that time. We adopted several recruitment methods, including via social media, personal contacts, the COVID-19 Social Study newsletter and website, national homelessness charities, and local services working with PEH. All participants needed to be aged 18 or over and living in the UK to take part. In addition to these criteria, people who were homeless during the COVID-19 pandemic and service providers who worked directly with PEH during the pandemic were sought. Members of the research team provided eligible participants with details about the purpose of the research in writing and verbally, informed them that their involvement was voluntary, and provided an opportunity for questions to be asked about the study. All participants signed a consent form, and provided demographic details (e.g., age, gender, ethnicity, accommodation (PEH) or service type and job role (service providers)). This information was received via email from participants who were interviewed remotely, or ‘in person’ if interviewed face-to-face. Those who agreed to participate remotely signed a consent form electronically prior to the interview, and those interviewed in person signed a hard copy of the consent form. Recruitment to the study continued until the research team had spoken to individuals from a range of demographic backgrounds and until no new major themes were identified from participant accounts.

### 2.2. Data Collection

Interviews were undertaken between April 2021 and January 2022 and were between 5 and 80 min (mean: 40 min). Interviews followed a semi-structured format using a topic guide of questions (see Appendix A) informed by existing theories of behaviour change [22]; social networks and mental health [23]; and health, stress and coping [24]. All service providers and eight PEH were interviewed remotely, and 14 PEH were interviewed in person. Those interviewed remotely were advised to find a quiet space on their own to maintain confidentiality and were offered the choice of their interview being conducted via telephone or video call. Interviews carried out in person were held within either a support service (*n* = 4) or hostel where the interviewee resided (*n* = 10). Interviews took place in a ventilated room and the researcher followed social distancing guidelines. Interviewers had previous experience in conducting qualitative research with vulnerable populations (JD is a research fellow in public health and healthcare clinician; TM, AB, and AM are postdoctoral research fellows in social science and behavioural health). Participants were not known to the member of the research team who interviewed them. Introductory information about the research project and the research team were explained verbally before the interview commenced. All interviews were audio-recorded and transcribed verbatim by an external transcription company. All participants who completed an interview were provided with a GBP 10 shopping voucher to thank them for their time.

### 2.3. Data Analysis

We used a reflexive thematic analysis approach [25] to analyse the data. Data were managed using NVivo 12 Pro software. JD was the primary data coder who read and coded all transcripts line-by-line. An initial coding framework was developed a priori based on key concepts from the interview topic guide. The framework was then applied to the transcripts and updated with new codes in response to content encountered in the transcripts. To ensure the consistency of the coding approach in identifying pertinent topics, a second researcher (TM) double coded 10% (*n* = 4) of the transcripts and coding was then compared and discussed between the researchers. On completion of coding, preliminary themes were generated by JD. The coding framework and subsequent themes were presented to the research team for discussion and feedback. Themes were refined based on this feedback, finalised, and reported with supporting quotes from participants.

## 3. Results

### 3.1. Overview of Participants Interviewed

We recruited 33 people (22 PEH residing in England and 11 service providers working in homelessness services across England, Scotland, and Wales, UK). Participants who experienced homelessness during the pandemic resided in a variety of settings at the time of interview, including in hostels (*n* = 11), rented homes (but had experienced a form of homelessness during pandemic) (*n* = 5), with friends/family (*n* = 2), hotel/B&B/sheltered accommodation (*n* = 2), and no fixed abode (including sleeping on the street or in a vehicle) (*n* = 2). The service providers interviewed represented several different organizations and services, such as local authority, social enterprise, or charities relating to health or social support, the arts, and accommodation. PEH participants were aged 24–59 years (mean age 40.5 years) and service providers were aged 34–53 years (mean age 46 years). Across all interviewees there was an equal distribution of men and women, and ethnicity was predominantly white British (*n* = 13, 59% of PEH and *n* = 6, 54% of service providers). Among PEH there was a range of educational attainment, from not completing GCSEs or equivalent to postgraduate education. Participant demographics are presented in Table 1 (PEH) and Table 2 (service providers).

### 3.2. Themes

Five themes were identified (Table 3) which provided insight into how the COVID-19 pandemic and associated social distancing restrictions impacted the health, wellbeing, and provision of services for PEH: (i) Understanding and adhering to COVID guidelines and restrictions; (ii) Mixed experiences of the ‘Everyone in’ initiative; (iii) Impact of social distancing guidelines on experiences in public spaces; (iv) Social support and connectedness to others; and (v) Adaptations to homeless service provision.

#### 3.2.1. Understanding and Adhering to COVID Guidelines and Restrictions

##### Challenges with Adherence to Changing Public Health Guidance

People who experienced homelessness during the pandemic and the staff who supported them acknowledged confusion around public health messaging and challenges with adherence. Adherence was often difficult due to limited access to the resources required to follow guidance (for example face masks, hand washing facilities, or hand sanitizer) or spending time in environments which were not conducive to social distancing.


*“They [people experiencing homelessness] said… they didn’t even have access to face masks and things at that point [first lockdown]. Washing facilities, once they were in the B&Bs and things, they were lucky if they had a sink in their room, but most things were communal, because we haven’t got big hotels in [location].”*
ID356 Female Service Provider, Wales

##### Difficulties of Restrictions in Communal Living Settings

There was also a clear sense amongst PEH that there was substantial difficulty for hostel staff to enforce public health guidance. Reasons were identified as possible lack of knowledge amongst staff, or residents not adhering to the advice given.


*“[In] the emergency accommodation, I think the biggest challenge was staff, to be honest… with the emergency accommodation service we were basically building a 120 bed complex needs hostel within a week, and staffing [it] with people who had never worked with homelessness before. So, you can just picture the scenes of what happens following that… We tried to pull together to as kind of a core of experienced workers, but it just meant that every single process was so much more challenging because it needed to be really simplified…it meant that when we were… starting to accommodate people, the kind of behaviour (sic) that might be common place in a hostel, like being aggressive or signs around addiction issues and lots of things like that, our staff members … sometimes became really hostile and kind of try and manage it in a punitive way and… all those situations, it could have made it a lot worse.”*
ID348 Male Service Provider, England

This situation proved stressful for many people living in shared accommodation facilities, particularly when sharing with people they did not know, and who therefore did not feel comfortable challenging other people’s behaviour.


*“It was hard, to be honest, because we were in a hostel of over 300 people and we had all kinds of people, there was no safety guidelines within the hostel, so I was kind of stuck indoors, the only time I would try and go out for a bit of air and whatnot would be night time, so yes that was quite hard to do, to follow- you know when they relaxed the restrictions, that kind of made me anxious, actually, to even go back out.”*
ID416 Woman, 40s, renting at time of interview, but lived in hostel during pandemic

There were varied explanations given by the PEH interviewed as to why some would not follow the social distancing guidance. A belief of societal rules not applying to them was described.


*“We live outside the society. We don’t feel part of the society. So, you know, who’s going to tell us to stay in lockdown?”*
ID277 Man, 40s, sofa surfing

Others shared a sense that their life was already full of risk and avoiding infection from COVID-19 was not the greatest risk or priority for them.


*“No one followed it. No, none of us did. I used to go back to the hostel and wash my clothes… I’d go in, see the girls, and wash my clothes... And no one followed social distancing. The rule was basically, we’re all sex workers, of course we’re not going to follow social distancing. Who gives a shit? And we were all drug users. We all shared drugs. We’re not going to do it two meters apart, especially when you’re cracking up a rock. The last thing you want to do is be two meters away from it.”*
ID308 Woman, 20s, renting at time of interview, but lived in hostel during pandemic

#### 3.2.2. Mixed Experiences of the ‘Everyone In’ Initiative

##### Accommodation Changes Due to the Pandemic

Both staff and PEH had mixed views about the government policies designed to protect PEH during the pandemic through protection from eviction and emergency accommodation provision. More specifically, there were positives and negative experiences of the ‘Everyone In’ scheme. Many PEH were grateful for this support, recognizing that this was not accommodation they would have been able to pay for themselves.


*“[on the streets] it was quite worrying, quite daunting, and I would have been in trouble if there was no-one to help me and I would have to stay on the street alone like that, but I got picked up by [charity] and I got housed… it was a treat, in a way, for us because most people got a double room, shower, they have to pay £100 a night and that’s not no money I’ve got and things I can do. So, it was sort of like a holiday for me.”*
ID388 Man, 40s, living in hostel

Many were positive about people feeling cared for, feeling glad to be off the streets, and recognizing that secure accommodation provided stability, which positively impacted their health and wellbeing.


*“The ability just to walk from a bed to a shower, and have laundry done free of charge every week... All of those kind of little things, made a huge, huge, huge difference. And, you know, I lost weight, my tension level, I didn’t actually have my blood pressure taken, but I’m pretty sure it would have come down…”*
ID274 Man, 50s, living in hotel

##### Hostel Life through Pandemic

However, there was also a recognition that for many people, hostel provision did not adequately meet their specific needs or the needs of their families.


*“I mean at the time he [interviewee’s son] was still under 18 so yes, we were housed together he was in the room, and he had to share the bed with me. It was overcrowded… sometimes we couldn’t stand each other… we would snap at each other sometimes, you know cooped up in a little room, no window to open, so it wasn’t the greatest experience.”*
ID416 Woman, 40s, renting at time of interview, but lived in hostel during pandemic

There were many examples provided of chaotic environments, a lack of food, and feeling unsafe or unhappy.


*“It’s never safe, is it? The hostel wasn’t safe. People used to bring stuff in. People used to bring people in. And you used to have to lock your door when you went for a piss in the night, because you thought someone would rob your stuff. And people would rob your stuff. They’d just go in and, quickly clean your room out, and leave.”*
ID308 Woman, 20s, renting at time of interview, but lived in hostel during pandemic

##### Pros and Cons of ‘Everyone In’

Moreover, for many participants there was a lack of choice over accommodation options, with some people being moved out of their local area and away from their support networks.


*“they have no choice… where they were put, so quite often you had people from very rural areas that were displaced and put into … the five major towns that they had no social links to… Some were impacted in that the children had to stay somewhere else… the place that they were taken to wasn’t suitable for children. So, people were actually picking between a home and their families, they did sort of turn down help at one point.”*
ID356 Female Service Provider, Wales

However, the process and success of ‘Everyone In’ resulted in many people—who otherwise may have been hidden from services by their homelessness—being housed together in locations that allowed services easier in-reach, with a net result of improved care co-ordination.


*“…they were then very eager for us to recruit new posts and get people into the service as soon as possible. You could have these specialist roles to go into the hostels and engage with these people because I think the council recognised that this was a real opportunity for them which was accidental in some ways, because if it wasn’t for the pandemic they wouldn’t have housed them all in the same place and they wouldn’t have sent these teams out there to sweep people off the streets and put them in somewhere.”*
ID329 Male Service Provider, England

#### 3.2.3. Impact of Social Distancing Guidelines on Experiences in Public Spaces

##### Increased Stigma from the Public

For many PEH, spending substantial time in public spaces is a usual part of their life. Many people we interviewed acknowledged that experiencing stigma was not uncommon before the pandemic, but there was a perception that the pandemic had escalated this. Some participants described noticing subtle differences in the way people interacted with them:


*“I was sat there begging outside the bank… it’s all quiet, becoming like a ghost town… wasn’t it? But I was sat there begging one day and people, they put the money a little bit over there.—just a little bit over there, not in my hand.”*
ID309 Woman, 40s, living in hostel

Whereas others described profound and frightening examples of stigmatization.


*“People just treated you like shit, especially during the pandemic. They just treated you like some sort of vector of disease. People were filming us, and putting us online, harassing us, violence went up. Yes, it was awful. It made things worse.”*
ID308 Woman, 20s, renting at time of interview, but lived in hostel during pandemic

Notably, of the people we interviewed who described experience of stigma (*n* = 6) far more were women (*n* = 5) than men (*n* = 1).

##### The Loss of Usual Survival Strategies

Interviewees referred to how the sudden societal changes imposed in response to the pandemic (closure of non-essential shops, remote access to services and the public being encouraged to work from home) impacted on how easily PEH could function in urban spaces. Many described that the usual ‘survival strategies’ they would adopt to access food, resources, or money were suddenly unavailable. Some described how begging was impacted by people no longer going to public spaces.


*“Obviously you couldn’t go and make money, you know, you couldn’t go and earn money and that. You could not do that because there was no one about. I just beg really. Or I do a little bit of work here and there. But I don’t do crime. Because there was no shops, there was nothing, you know what I mean?”*
ID277, Man, 40s, sofa surfing

Others described the impact closure of services had on the ability to access free food or showers.


*“Well in [city], there’s not a short supply of food -so you’re always, you wouldn’t go hungry in [city]. Yes, people just give you food or hand-outs or whatever they call them. Well, there was no services open. The day centres and that where you can go and get showers and stuff, they were shut.”*
ID387 Man, 20s, living in hostel

In addition, usual survival strategies, such a shoplifting, were impacted due to social distancing measures imposed on shops.


*“The shops were limited, what you could get was limited… And I found it hard because a lot of obviously, the food shops then upped their security, didn’t they? So, it did get a little bit harder… I tend to kind of like slip things into my bag under the cover of, you know, people being in front of me and stuff like that, you know? But not having people, so many people in the shops it was harder for me to do that, yes.”*
ID301 Woman, 50s, living in hostel

#### 3.2.4. Social Support and Connectedness to Others

##### Experience of Limited Social Networks

Social support and connectedness were discussed by many participants. Those who usually relied on friends for support found that their networks diminished as people who would usually be geographically close moved away during lockdowns.


*“In terms of having friends who could do anything [for me], that’s near-on impossible now. Urm, partly because most of my friends are out of town now. Usually they’re central [city] but because of the pandemic, they went back to friends or family outside of [city]. So yeah, when I crash, there’s me going like, ah, I hope this doesn’t last too long.”*
ID274 Man, 50s, living in hotel

For some, this lack of social support was attributed to a deterioration in managing addiction issues during the pandemic.


*“No, my family—well, I don’t speak to them anymore… I see [support agency] as well as part of… probation… So, I was coming down off the script and then—And, yes, no support or whatever, so I just ended up using.”*
ID271, Man, 30s, living in hostel

Some participants described limited friendships or support networks prior to the pandemic, which in turn may have provided a degree of resilience during times spent alone in lockdowns.


*“I have a large circle of acquaintances. But I enjoy being on my own more than anything, though.”*
ID312, Woman, 30s, living in hostel

##### Opportunities to Reconnect with Family and Friends

By contrast, some participants were able to access technology through the pandemic and made greater connections with family and friends. This may, in part, have been due to the people they made contact with having more time for phone/video calls due to their own lack of opportunity for ‘usual’ work or social lives.


*“I have better connections with some of my friends who are far away, so they do calls and Zoom calls because everybody… has been mostly indoors and lots of my employed friends have been at a loose end. So, I have built up better connections with quite a few people because of that.”*
ID232 Woman, 40s, living in vehicle

Additionally, for some the lockdowns provided a catalyst for improving or rekindling social connections, which in turn acted as motivation to improve health.


*“And I think my son, as well, is a big part of it. Even though I was using [drugs] after I had him and everything, I think it was, just getting on… getting on after about four years have gone by, I said to myself, ‘I’ve missed a lot of his life, from doing all this.’ And I started getting in contact with him. It was mainly him, as well, from talking to him every other day… But now, when lockdown comes, that phoning him regularly and that, it kind of made me not want to use [drugs], because I knew I wanted to meet him, try to meet him.”*
ID310 Man, 20s, living in rented house at time of interview, but homeless during pandemic

#### 3.2.5. Adaptations to Homeless Service Provision

##### Loss of Formal Social Support Services

Interviewees noted the profound impact of loss of access to usual formal support services, such as drop-in centres and soup kitchens commonly accessed by PEH.


*“So, all day centres were closed, all places where they would go to get a free meal or even just… for those who were on the street… a place to get a shower. All of a sudden, none of it was available. Some people were really reliant on just simple things like having a plug socket to be able to charge their mobile phone, no longer having that and then they couldn’t use a phone to be able to call for support either.”*
ID348 Male Service Provider, England

It was clear that for PEH the loss of much in-person service provision and adjustment to remote access also proved challenging, with some interviewees describing the negative impact on their wellbeing and ability to maintain positive connections.


*“Having the phone call appointments... it sort of made them seem like they’re further away… you know what I mean? So, there’s less support.”*
ID271 Man, 30s, living in hostel


*“I felt incredibly frustrated, because I couldn’t see people. I couldn’t see my worker. And I hated that, because I was moving away from drugs, and therefore moving away from drug users. But then I wasn’t replacing that with anything. I was trying to move away from the drug life... But then it just made it more difficult.”*
ID308 Woman, 20s, renting at time of interview, but lived in hostel during pandemic

##### Digital Exclusion

PEH and service providers noted that digital exclusion was a huge challenge amongst PEH, whether due to lack of technology (phones, tablets, laptops) or a lack of internet access (no data on phones, no access to wifi, or an inability to charge devices).


*“On the access to services… a massive impact. For everybody to try and contact services and things, relevant services, has been difficult, let alone if you haven’t got the resources such as the internet, phones, computer, everything like that… There was no face-to-face opportunities, people felt very discarded.”*
ID356 Female Service Provider, Wales

For others, a lack of understanding around how to use the technology required to access remote services was the primary barrier.


*“I’m not the best due to, technology and academic, due to being kicked out at 16, not being able to concentrate properly and then a lot of prison. So, I didn’t learn too much about or have the practice about, so I do need a key worker usually, to help me do stuff. I remember trying to get my medication and a sick note … because I was staying at my auntie’s before I went into… that backpacker’s hostel and it was difficulter, harder than normal”*
ID388, Man, 40s, living in hostel

One PEH illustrated difficulties in accessing services remotely, specifically highlighting the fragility of connecting solely by telephone when personal phone numbers can get disconnected or change.


*“She phoned me once a week and that. I just lost contact. I think she went off and then my number got changed because of something. But I imported my old number, so, within them three days… I think it was in them three days that she was meant to ring me. And then it just stopped.”*
ID310, Man, 20s, living in rented house at time of interview, but homeless during pandemic

##### Benefits and Challenges of Services Operating Remotely

There were substantial benefits described by service providers about service moving online. Some felt that once PEH were set up with the correct technology for remote consultation, there was improved connection between service providers and users. Service providers observed some of their female service users experiencing remote access more positively than in-person service provision.


*“… some women preferred it because they weren’t coming into a room full of men… And for vulnerable females with complex needs, particularly those who possibly have been sex workers before, they sometimes come into… group activities and there would be men there, who they knew from the street, who had done things to them in the past, whether that was exploiting them or preying on them or paying them for sex and things and they didn’t particularly want to be sitting in a group with that person talking about very personal and private things… they found it easier to engage… online, which I think was really fascinating. But [that is] a positive that’s come out of the pandemic and something that’s gonna keep going. So, even as we come back into normal treatments, we’re still gonna be offering online interventions and online groups for as long as we can so people have an option of the kind of hybrid way of engaging with the service.”*
ID329 Male Service Provider, England

Additionally, some service providers could see real benefits in the flexibility that combining working remotely and in person care allowed.


*“People working from home, they can focus on their caseload, focus on their clients, focus on what they’ve got to do and actually get stuff done. And so, I think a hybrid way of working is definitely gonna be the way forward… And we’ve achieved things that we probably wouldn’t have thought would be possible if we hadn’t been forced to do it, so in that sense it’s been positive… When we initially assess someone, we need to see them in the flesh. But going forward, once we’ve assessed their risk, if they are low risk and its preferable to them, there’s no reason why we can’t engage with them partially in a virtual way if that’s working better for them. So, I think we’ll have a more flexible adaptable more tailored series of interventions… than we used to.”*
ID328 Female Service Provider, England

However, some service providers held mixed views on the move to remote working for services that support PEH. Some felt that it was more difficult to make an accurate assessment of people’s wellbeing when consulting over the phone.


*“If you walked in to see me today, I would do, you know, a visual assessment of you. And people, I guess, that are declining in health might not see it themselves, that they’ve lost weight, they have yellowing, they’ve got ascites to the tummy, whatever it might be. Their injection sites are smelly, you know, they might not know. And that’s one thing that I think we’ve missed, is eyes-on assessment.”*
ID331 Female Service Provider, England

##### Impact of Changes to Service Provision on Staff

Service providers highlighted the pressure that working with homeless and vulnerably housed people through the pandemic imposed on them, whether that was increased service demand, inefficiencies caused through balancing in person and remote access, staff burn out, or the emotional toll of dealing with people in crisis.


*“My diary’s a mess, I spent a few weeks just staring at my diary, just going, I can’t do this, I can’t. It’s impossible. Looking at our case notes going up and up and up… A lot of people coming back who dropped out of the service system have then come back, so we’ve got a revolving door going on. I’m exhausted. I’m actually at a point where I’m thinking I don’t think I can do this work anymore; I just can’t do it. Just overwhelmed. I think we’re all overwhelmed... So, feel like what my job has become, is putting people in other people’s waiting lists, and then dealing with their stress while they’re on that waiting list. That’s hard.”*
ID333 Female Service Provider, Scotland

However, as well as service changes brought in during the pandemic being of benefit to some individuals, one service provider noted how whole services were working together more positively and proactively at this time and that this experience had real potential to alter how homelessness services collaborated beyond the pandemic.


*“Because of the partnership work that was going on from the health providers and the commissioners, the needs of the residents were met pretty quickly... It was a great example of what the sector should be capable of doing, and for the first three months, I was like, this could be a really promising opportunity for the homeless sector to start doing things differently and really see the benefits of genuine joined-up working, where you’re not competing for contracts, you’re not competing for who gets the kudos.”*
ID348 Male Service Provider, England

Staff were also positive about how the pandemic highlighted the problem of homelessness, which in turn fuelled improved partnership working and efforts to support people in housing crisis.


*“…we did have an outreach and engagement team and we did a lot of joint work with the police and with… a homeless charity and the anti-social behaviour team and teams from the council trying to engage with people… bringing them in to treatment… that existed pre-COVID and it got a real boost actually during COVID because I think, you know, the crisis became very visible and the council had to do something about it.”*
ID329 Male Service Provider, England

## 4. Discussion

This research investigated the impact of the COVID-19 pandemic and associated societal restrictions on the health and well-being of people experiencing homelessness during that time. It exposed issues with the understanding of and adherence to public health guidance and restrictions; the experience of people accommodated by the ‘Everyone In’ initiative; the impact of social distancing guidelines on PEH experiences in public spaces; the importance of social support and connections to others; and, how homelessness services adapted their provision. The themes our study identified add to the existing (limited) literature which reports the experiences of PEH during the pandemic and their access to accommodation and support services. By bringing together the views of PEH and the staff that supported them we offer a depth of insight that previous work has not provided.

Our theme regarding “understanding and adhering to COVID-19 guidance” showed variation that was often influenced by the context in which the individual found themselves. For example, those in hostel settings described how these environments were often not conducive to social distancing, due to limited access to the resources required to follow guidance. So, despite concerns of COVID-19 transmission, the capacity to adhere to guidelines within hostels and other communal living environments was limited. Moreover, our research suggests fear of transmission was confounded by perceptions PEH had that staff struggled to enforcing public health guidance within hostels. Other research carried out during the pandemic with this group reported staff in emergency accommodation being unhelpful or having a ‘bad attitude’ [26]. The combination of limited resources and staff difficulties enforcing guidance within accommodation facilities heightened stress amongst the PEH we interviewed. Indeed, feeling unsafe and vulnerable to infection has been found to be predictive of poorer mental health among other population groups: those with long-term health conditions, for example, reported high levels of fear and anxiety related to the consequences of COVID-19 infection [27]. Our research suggests that during future infectious disease scenarios, additional support is needed to attend to the vulnerabilities affecting PEH, including appropriate accommodation and staffing of these facilities. This finding fits with the concept of proportional universalism, where health actions must be provided on a scale and intensity that is proportionate to the level of disadvantage and need [28].

Under the theme of “mixed experiences of the ‘Everyone in’ initiative” although many reported the initiative to be well received, some participants described unsuitable accommodation settings and little control over where they were offered housing. Some described being moved out of their local area and away from support networks, or into small hostel rooms shared with family members. Housing providers and support services should recognize the implications of imposing a lack of choice on people who need accommodation during a public health emergency and seek to find ways to serve the needs of those most affected. This includes tailored support and the need to ensure a variety of housing options to prevent repeat cycles of homelessness and to support long-term housing needs of the diverse needs of PEH [29]. Qualitative research carried out in the summer of 2021 suggested that when funding of the ‘Everyone In’ initiative ceased, people had moved to a range of accommodation settings (hotels, flats, hostels, and the street). In a relatively short space of time many had been moved more than once and all described feeling concerned and anxious about their accommodation. This research suggests that as COVID-19 restrictions lifted in the UK, housing support needs of PEH were not being met [30].

Our data illustrates how the pandemic posed several challenges beyond the usual difficulties associated with homelessness, in particular within the theme of “impact of social distancing guidelines affected PEH in public spaces”. This included challenges with accessing facilities for hygiene and self-care, food and making money, as adoption of ‘survival strategies’ that PEH usually relied on became harder or not possible, thus increasing their vulnerability further. The theme of “social support and connectedness to others” showed that these challenges were felt particularly acutely in the absence of traditional support networks (e.g., friends and family). During future infectious disease scenarios, it is important that PEH are provided with sufficient economic and auxiliary support (including food) to buffer gaps left by limitations to ‘survival strategies’ usually relied upon. In making appropriate provision, this could serve to limit the increased risk of criminalisation and stigmatisation in this population, that can be associated with adoption of survival strategies [31]. The provision of ongoing peer support should also be prioritized given the positive role of social networks and linkages in fostering positive health outcomes and greater engagement with services [32].

Relatedly, our research indicated that public hostility towards PEH increased during the pandemic. This was often linked to perceptions of being infected or ‘unclean’ (rather than actual infection), as has been reported by other population groups during the pandemic, including healthcare workers [33]. PEH are already a highly stigmatised group, the effects of which have been shown to affect feelings of self-worth, operate as a barrier to healthcare engagement [34,35], and increase symptoms of mental ill-health [36]. Although difficult to address, community and peer-led advocacy groups have been instrumental in forming and empowering collective identities among similar populations, including people who use drugs [37] and sex workers [38], thereby increasing self-esteem, ameliorating internalised stigma and increasing agency [36]. Such interventions are critical in the absence of formal support or protection for these groups, particularly during the pandemic. More specifically, our research highlighted that women were more likely to describe experiences of stigma than men and that some women reported preferences for remote rather than in-person service provision. Pre-pandemic research has shown that women experience homelessness differently than males and that they have worse mental health and report greater adverse childhood experiences [39]. Pathways into homelessness also differ and have been linked to domestic abuse and economic marginalization [40]. Our work adds to this literature and suggests a distinct need to provide women-centred, trauma-informed homelessness support and services.

Finally, the theme of “adaptations to homeless service provision” highlighted that whilst remote service provision, such as the use of telephone or video communication to replace in-person appointment, were implemented rapidly, there was substantial disruptions to care. These mostly concerned the use of remote or digital psychosocial support or assessment, with issues of acceptance, accessibility, and level of care reported by both clients and providers. The suspension of some face-to-face services—including peer support groups—were found to exacerbate mental health symptoms among some participants. This finding corroborates research that explored access to mental health and substance use support for PEH [17]. In future emergencies, there remains a need to ensure access to telehealth services are fully accessible to PEH, as well as learning from services that remained open during this period. This is particularly important given how outreach based provision are best placed to tackle key barriers to service access among PEH [41].

### 4.1. Implications for Practice

Our research highlights that policy makers and those involved in public health communication must learn from people with experience of homelessness to maximise effectiveness of future public health strategies and communication. Additionally, housing providers and support services should recognize the implications of imposing a lack of choice on people who need accommodation during a public health emergency. Finally, it is essential to recognize that the sudden loss of usual strategies and support was destabilising for people experiencing homelessness, increasing vulnerabilities in an already marginalised and precarious group, which may negatively influence their health.

### 4.2. Strengths and Limitations of Research

A strength of this work is that it gathered opinions from a broad spectrum of people who experienced homelessness during the pandemic, including people who had lived in hostels, on the streets, in vehicles, in emergency hotels, and with family and friends as well as a diverse demographic sample. It also gathered the perspectives of a variety of people working to provide support services for PEH throughout the pandemic. The triangulation of these people’s experiences is powerful and provides in-depth insight from multiple perspectives. Another strength of this work was that once social distancing restrictions were eased in the UK, the research team was able to conduct in-person interviews, thus enabling greater reach to people who would not otherwise have been able to speak to us, due to digital exclusion.

However, the limitations of this work must also be acknowledged. Data collection occurred between April 2021 and January 2022 and coincided with differing levels of restrictions but did not include periods of lockdown in UK. Participants were therefore retrospectively describing their experiences of the ‘Everyone In’ scheme, which for some did not reflect their housing status at the time of the interview. People were being asked to reflect on their experiences over the previous 1–1.5 years and recall of events may have been influenced by more recent events and changes in circumstance. Early data collection was limited to online interviews, which may have excluded some people who did not have access to digital technology, thus failing to represent all perspectives during this period. Also, data collection for this study ceased before the onset of the cost-of-living crisis in the UK, which may mean experiences in the pandemic after January 2022 have differed.

## 5. Conclusions

This research highlights successes and difficulties in supporting PEH during the COVID-19 pandemic and informs planning for similar public health events. Although the ‘Everyone in’ scheme was valued by many, it was also characterised by a lack of choice or control amongst people being ‘brought in’. Housing providers and support services should recognize the implications of imposing a lack of choice on people who need accommodation during a public health emergency. The loss of usual support was also destabilizing for PEH and was exacerbated further by not being able to easily rely on usual ‘survival tactics’, which may negatively influence their health. An application of proportional universalism in the provision of additional supports is therefore crucial to prevent further health decline of an already vulnerable population. In future infectious disease scenarios, it is also important that policy makers and those involved in public health communication continue to learn from people with experience of homelessness to maximise effectiveness of public health strategies and communication.

## Figures and Tables

**Table 1 ijerph-19-15526-t001:** Characteristics and demographics of PEH interviewed.

	Demographics	Range/*n* (Mean)
Age		24–59 years (40.5 years)
Gender	men	11
	women	11
Ethnicity/ multi-ethnic group	white British	13
	white other	3
	mixed	3
	black or black British	2
	Asian	1
Housing situation	hostel	11
	rented (but homeless during pandemic)	5
	hotel/B&B/sheltered	2
	friends/family	2
	no fixed abode	2
Highest educational qualification	did not complete GCSE or equivalent	7
	GCSE or equivalent	6
	post-16 vocational course	3
	A-levels or equivalent	2
	undergraduate	3
	postgraduate	1
Employment status	unable to work due to ill-health	13
	unemployed	5
	part time employment	2
	full time employed	2

**Table 2 ijerph-19-15526-t002:** Characteristics and demographics of service providers interviewed.

	Demographics	Range/*n* (Mean)
Age		34–53 years (46 years)
Gender	men	5
	women	6
Ethnicity	white British	6
	white other	4
	black British	1
Location	England	9
	Scotland	1
	Wales	1
Experience working with PEH	1–5 years	5
	6–10 years	2
	>10 years Not stated	3 1

**Table 3 ijerph-19-15526-t003:** Themes relating to the impact of the COVID-19 pandemic and associated restrictions on people experiencing homelessness.

Theme	Sub Theme
Understanding and adhering to COVID guidelines and restrictions	Challenges with adherence to changing public health guidance
	Difficulties of restrictions in communal living settings
Mixed experiences of the ‘Everyone in’ * initiative	Accommodation changes due to the pandemic
	Hostel life through the pandemic
	Pros and cons of ‘Everyone In’
Impact of social distancing guidelines on experiences in public spaces	Increased stigma from the public
	The loss of usual survival strategies
Social support and connectedness to others	Experience of limited social networks
	Opportunities to reconnect with family and friends
Adaptations to homeless service provision	Loss of formal social support services
	Digital exclusion
	Benefits and challenges of services operating remotely
	Impact of changes to service provision on staff

* ‘Everyone In’ was a UK government public health strategy to provide emergency accommodation to people experiencing homelessness early in the COVID-19 pandemic.

## Data Availability

The data are not publicly available due to their containing information that could compromise the privacy of research participants. The interview topic guide is available in Appendix A.

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
