# Peer review of "The Impact of the COVID-19 Pandemic and Associated Societal Restrictions on People Experiencing Homelessness (PEH): A Qualitative Interview Study with PEH and Service Providers in the UK"

_ijerph, 2022, doi:10.3390/ijerph192315526_

Round 1

Reviewer 1 Report

This is a very significant article, addressing a humanistic issue for people who live in the margins of society. While the focus during lockdowns and ‘stay home’ campaign was on those having a secure shelter over their head, homeless people were somehow left behind.
This study provides evidence that social interventions worked quite effectively in order to address the needs of PEH.

The article is well written and made very good use of NVivo 12 Pro software.

The discussion section is also very well written and presents in full detail all the findings, strengths and limitations.

Author Response

Thank you for the positive comments about our article. We are delighted you consider our manuscript fit for publication without amendments. 

Reviewer 2 Report

Dear Authors,

This paper shows an interesting area of research with reference to the effects of social restrictions due to coping with pandemic COVID 19.

Personally, I found the paper lacking in many aspects, some of detail others substantial. Listed below are the elements that, in my opinion, make the paper currently unpublishable:

The title

Why do you talk about the effects of the pandemic in the title? You actually investigated the effects of social restrictions during the pandemic. The World Health Organization speaks of direct psychic effects such as fear of virus infection and indirect effects such as those due to social restrictions and infodemic.

Results

why do you describe ethnicity of the sample with the categories: white British,  white other,  mixed,  black or black British, Asian. The categories you have used are not ethnicities but multi-ethnic groups. In my opinion, to distinguish the ethnic category by skin colour  is not correct.

Why did you not distinguish the results obtained from the interviews with homeless (22) with those of the  service providers working in homelessness services (11)? Thus you risk mixing very different aspects together

Honestly, it is difficult to evaluate a content analysis survey without any data. You have reported only examples. There is no table where the codes found, a tally of codes or families of codes are reported. Your comments are very free and completely disengaged from what you observed, at least for the reader it is impossible to understand how you arrived at such conclusions. Any qualitative method  needs an "anchor" to an observed datum, otherwise they are simply free deductions. It is necessary to list all the codes and the frequency of each in a table.

Your conclusions suffer from speculation unsupported by any verified data; many of the results reported seem to be general conclusions.

In summary, for the paper to be considered for publication, more systematic work is needed:

-Results need to be differentiated between homelessness and homeless support association workers.

-Tables where codes and frequency are reported.

-In discussions, do not report generic expressions such as, "Our data illustrate how the pandemic..." without showing any link between data drawn from interviews and conclusions.

I personally feel that the work needs substantial revision.

Sincerely

Author Response

Please see attached file for detailed responses to to reviewer 2's comments.

Reviewer 3 Report

·        Summary

The manuscript addresses one of the paradoxes associated with the CoVid-19 pandemic, i.e., the recommendation of ‘staying at home’ for those who, in fact, do not have it. The case of the United Kingdom is considered, and, in methodological terms, 33 semi-structured qualitative interviews were used.

·        Broad comments

From time to time -- but very rarely, in fact -- I have the opportunity to review a manuscript which, due to its content and form, I cannot, through my recommendations, help to improve. This is the case of this manuscript because, in my opinion, it meets, in its current version, all the conditions to be published.

Still, I dare to share with the authors a concern, i.e., whether as a result of this pandemic there will be an increase in the number of homeless people.

Author Response

We would like to that thank reviewer 3 for their generous and supportive remarks. We are delighted they consider our manuscript fit for publication without amendments. 
